# Lipopolymers as the Basis of Non-Viral Delivery of Therapeutic siRNA Nanoparticles in a Leukemia (MOLM-13) Model

**DOI:** 10.3390/biom15010115

**Published:** 2025-01-13

**Authors:** Panadda Yotsomnuk, Amarnath Praphakar Rajendran, Daniel Nisakar Meenakshi Sundaram, Luis Carlos Morales, Cezary Kucharski, Mohammad Nasrullah, Wanwisa Skolpap, Xiaoyan Jiang, Spencer B. Gibson, Joseph Brandwein, Hasan Uludağ

**Affiliations:** 1Department of Chemical and Materials Engineering, Faculty of Engineering, University of Alberta, Edmonton, AB T6G 1R1, Canadamnasrull@ualberta.ca (M.N.); 2Department of Chemical Engineering, Faculty of Engineering, Thammasat University, Pathumthani 12120, Thailand; 3Faculty of Pharmacy and Pharmaceutical Sciences, University of Alberta, Edmonton, AB T6G 2H1, Canada; 4Department of Medical Genetics, Terry Fox Laboratory, British Columbia Cancer Research Institute, University of British Columbia, Vancouver, BC V6T 1Z3, Canada; 5Department of Oncology, Faculty of Medicine & Dentistry, University of Alberta, Edmonton, AB T6G 1Z2, Canada; 6Division of Hematology, Department of Medicine, Faculty of Medicine & Dentistry, University of Alberta, Edmonton, AB T6G 2B7, Canada

**Keywords:** lipopolymers, siRNA therapy, nanoparticle, siRNA polyplex, MOLM-13, FLT3, KMT2A::MLLT3

## Abstract

Small interfering RNA (siRNA) therapy in acute myeloid leukemia (AML) is a promising strategy as the siRNA molecule can specifically target proteins involved in abnormal cell proliferation. The development of a clinically applicable method for delivering siRNA molecules is imperative due to the challenges involved in effectively delivering the siRNA into cells. We investigated the delivery of siRNA to AML MOLM-13 cells with the use of two lipid-substituted polyethyleneimines (PEIs), a commercially available reagent (Prime-Fect) and a recently reported reagent with improved lipid substitution (PEI1.2k-PHPA-Lin9). The siRNAs utilized in this study were targeting the oncogenes FLT3 and KMT2A::MLLT3. Both lipopolymers gave similar-size siRNA complexes (210–220 nm) with positive *ζ*-potentials (+17 to +25 mV). While the binding efficiency of both lipopolymers to siRNA were similar, PEI1.2k-PHPA-Lin9 complexes were more resistant to heparin-induced dissociation. The quantitative analysis of gene silencing performed by qPCR as well as immunostaining/flow cytometry indicated significant reduction in both FLT3 expression and FLT3 protein after specific siRNA delivery. The desired inhibition of cell growth was attained with both FLT3 and KMT2A::MLLT3 siRNAs, and the combination provided more potent effects in both cell growth and colony formation assays. Induction of apoptosis was confirmed after specific siRNA treatments using the Annexin V assay. Using Luc(+) MOLM-13 cells, the growth of the xenografted cells was shown to be retarded with Prime-Fect-delivered FLT3 siRNA, unlike the siRNA delivered with PEI1.2k-PHPA-Lin9. These results demonstrate the potential of designed lipopolymers in implementing RNAi (via delivery of siRNA) for inhibition of leukemia growth and provide evidence for the feasibility of targeting different oncogenes using siRNA-mediated therapy.

## 1. Introduction

Acute myeloid leukemia (AML) is a clonal malignancy that disrupts normal myeloid differentiation, causing the accumulation of immature progenitors in bone marrow and subsequent failure of hematopoiesis [1]. AML treatment has been challenging, with relatively low 5-year survival rates of approximately 5–15% for older patients and 35–40% for younger patients or patients less than 60 years of age [2,3]. The standard induction therapy for AML patients still includes cytarabine and an anthracycline (e.g., daunorubicin) as the backbone. In childhood AML, approximately 60% of patients can achieve a cure through the use of highly intensive chemotherapy [4]. Chemotherapy resistance is the main contributor to the extremely low overall survival rates in some patient groups. Current chemotherapy is frequently incapable of completely eradicating the leukemia-initiating cells since they have a quiescent nature, remain unresponsive to chemotherapy, and cause disease relapse under suitable conditions [5]. Oncogenes that derive the leukemia-initiating cells towards uncontrolled proliferation are the underlying basis of the disease, and new therapies that eliminate the oncogenes are urgently needed to improve patient survival.

RNA interference (RNAi) is a regulatory mechanism for gene expression that can be therapeutically employed to curb uncontrolled cell proliferation by targeting oncogenes. It utilizes small regulatory RNAs, including microRNAs (miRNAs) and small interfering RNAs (siRNAs), to specifically silence or reduce the expression of target messenger RNAs (mRNAs) through sequence-specific interactions. After the discovery of RNAi in *Caenorhabditis elegans* [6] and implementation of siRNA in mammalian cells [7], RNAi has garnered considerable interest as a potential treatment for numerous cancers, particularly in cases where there is a lack of accessible or ‘druggable’ targets.

Although there have been successful applications of siRNA delivery targeting specific oncogenes in different malignancies [8,9,10], clinical use of siRNA therapies still faces considerable challenges. The approved siRNA drugs are generally effective in liver-centered diseases since this is the major organ for deposition of exogenous siRNA after systemic administration. The pharmacokinetic profile of naked siRNA is limited due to its degradation by RNase A in the extracellular environment and its inability to cross cellular membranes owing to its anionic properties [11,12,13]. Chemical modification of siRNAs offers an approach to enhance its stability in circulation. Alternatively, nanoparticulate systems derived from functional materials are required to protect siRNAs from degradation, while facilitating their passage across the membranes of target cells. For the intracellular delivery of anionic siRNA, cationic polymers have been shown to be safer carriers than viral vectors due to their lack of genomic integration. Furthermore, the polymers can be chemically engineered and adapted to specific requirements of the applications [14,15,16]. In this study, we propose the deployment of cationic lipopolymers for siRNA delivery and show the feasibility of this approach by silencing clinically relevant oncogenes in an AML model.

As a receptor tyrosine kinase, FMS-like tyrosine kinase 3 (FLT3) is a critical target for emerging therapies in AML and is commonly found in hematopoietic progenitor cells. It is notably elevated in a substantial number of AML blast cells [17]. The FLT3 internal tandem duplication (FLT3-ITD) leads to persistent activation of FLT3 signaling, consequently constitutively activating several downstream signaling pathways, and driving hematopoietic cells to proliferate independent of growth factors [18]. The anti-apoptotic and pro-proliferative functions of FLT3 mutations in AML cells indicate its potential as a molecular target, facilitating recent development of FLT3 protein tyrosine kinase inhibitors. Despite significant progress in drug discovery and insights into the molecular mechanisms associated with FLT3 mutations, less-than-desirable clinical response and drug resistance are still observed among some patient populations [19]. Another target for acute leukemias is the histone–lysine N-methyltransferase 2A (KMT2A) gene, previously referred to as the MLL (mixed-lineage leukemia or myeloid/lymphoid) gene, which can undergo translocations with different partner genes. Certain fusion genes are associated with distinct subtypes of leukemia such as KMT2A::AFF1 and KMT2A::MLLT3 (also known as MLL-AF4 and MLL-AF9, respectively). The t(9;11) chromosomal translocation results in the KMT2A::MLLT3 fusion gene, which is implicated in a subset of human acute monocytic leukemia and has the capability to transform hematopoietic progenitor cells [20,21,22]. During hematopoietic cell development, both the wild-type KMT2A and MLLT3 proteins function as integral components of protein complexes. The KMT2A, as an epigenetic regulator, contributes to the initiation of transcription, while MLLT3 is involved in the elongation phase of gene transcription by methylating specific histones, leading to activated promoters [23,24]. It is believed that the fusion protein KMT2A::MLLT3 combines both properties, leading to an enhanced activation of target genes through elongation and transcriptional initiation.

In this study, we targeted KMT2A fusions and constitutively activated FLT3 using siRNA nanoparticles formulated with polyethyleneimine (PEI)-derived lipopolymers. The cationic polymer PEI is extensively used in non-viral gene delivery applications because of its intrinsic ability to effectively encapsulate genetic material via electrostatic interactions [25]. The PEI is recognized for its strong H^+^-buffering capacity within the acidic endosome, which facilitates H^+^ binding and enhances endosomal osmotic pressure, which in turn causes the endosomal membrane to burst to release its contents [26]. However, the effective isoforms, high-molecular-weight (MW) PEIs, are generally linked to higher cytotoxicity [27,28]. On the contrary, low-MW PEIs are less toxic but also inefficient as gene delivery agents [29,30] and require functionalization to act as an effective carrier. We previously showed that hydrophobic lipids are ideal for PEI modification due to their endogenous nature [31,32,33]. Promising PEI derivatives were successfully developed by modifying it with various lipids [34,35], leading to improved uptake and transfection efficiency relative to the native PEIs. Here, we explore the feasibility of RNAi in AML models by employing siRNA/lipopolymer complexes to target the oncogenes FLT3 and KMT2A::MLLT3. The lipopolymers were synthesized from low-molecular-weight PEI (1.2 kDa) by chemically conjugating an optimal balance of hydrophobic (lipidic) groups to PEI [32,36]. We compared two lipid-modified PEIs, one where the lipid was grafted via a p-hydroxyphenylacetic acid (PHPA) linker and a commercially available lipopolymer (Prime-Fect), to evaluate their feasibility for treating leukemia with siRNAs. Relevant features of the siRNA complexes were comparatively characterized for both delivery vehicles to gain insight into their performance.

## 2. Materials and Methods

### 2.1. Materials

The PHPA-modified PEI with 4.8 lipids/PEI was previously described [36]. Briefly, the linolenic acid chloride was first reacted with PHPA overnight in acetone and purified, after which it was coupled to PEI via EDC/NHS activation for 24 hrs. The product was obtained by precipitation in ice-cold diethyl ether (3×). The transfection reagent Prime-Fect was purchased from RJH Bioscience Inc. (Edmonton, AB, Canada). RPMI 1640 medium with L-glutamine, phosphate-buffered saline (PBS), Hank’s balanced salt solution (HBSS), fetal bovine serum (FBS), and penicillin/streptomycin (10,000 U/mL and 10 mg/mL) were obtained from ThermoFisher Scientific (Ottawa, ON, Canada). MTT (Methylthiazolyldiphenyl-tetrazoluim bromide) and anhydrous dimethyl sulfoxide (DMSO) were obtained from Sigma-Aldrich (St. Louis, MO, USA). Human methylcellulose base media (Cat. No. HSC002) and human methylcellulose enriched media (Cat. No. HSC005) for the human colony-forming cell (CFC) assay were supplied by R&D systems, Inc. (Oakville, ON, Canada). *KMT2A::MLLT3* (ref. No. SO-3130160G, Dharmacon, Lafayette, CO, USA) and *FLT3* (ref. No.UA316322, BioSpring GmBH, Frankfurt, Germany) siRNAs were purchased from commercial sources. The sequences of these siRNAs were as follows: sense: 5′-CUUUAAGUCUGAACAACCCUU-3′/antisense: 5′-GGGUUGUUCAGACUUAAAGUU-3′ and sense: 5′-CUAGAGUUUACCCUCAAAGUU-3′/antisense: 5′-UUGAUCUCAAAUGGGAGUUUC-3′, respectively. The scrambled siRNA (C-siRNA) as a negative control (Cat. No. DS NC1) and 6-carboxyfluorescein (FAM)-labeled scrambled siRNA were supplied by IDT (Coralville, IA, USA). SYBR^®^ Green II was obtained from Cambrex BioScience (Rockland, MD, USA). The SensisFAST^™^ SYBR Hi-ROX kit and SensisFAST^™^ cDNA synthesis kit were obtained from Bioline (Memphis, TN, USA), and total RNA was isolated from the cells using TRIzol reagent (Invitrogen, Carlsbad, CA, USA). *FLT3* forward primer (5ʹ-AGG GCA ACT ACT TTG AGA TGA G-3ʹ) and reverse primer (5ʹ-AGT ATC CGG TGT CGT TTC TTG -3ʹ), bActin forward primer (5ʹ-CCA CCC CAC TTC TCT CTA AFF A-3ʹ) and reverse primer (5ʹ-AAT TTA CAC GAA AGC AAT GCT ATC A-3ʹ), and *KMT2A::MLLT3* forward primer (5ʹ-CTG AAT CCA AAC AGG CCA CCA CTC-3ʹ) and reverse primer (5ʹ-TCA CCA TTC TTT ATT TGC TTA TCA GA-3ʹ) were obtained from IDT (Coralville, IA, USA).

### 2.2. Cell Culture

Wild-type MOLM-13 and MV4-11 AML cell lines (originally obtained from ATCC and routinely passaged in the investigators’ labs) were maintained in RPMI medium supplemented with 10% FBS and penicillin/streptomycin, and were incubated at 37 °C with 5% CO_2_. Luciferase-positive (Luc+) MOLM-13 cells were a gift from Dr. Xiaoyan Jiang (Medical Genetics, University of British Columbia). The cells were subcultured after reaching 80% confluency and passaged at the original count concentration of 20%.

### 2.3. Preparation of siRNA Complex for Cell Delivery

The study design for siRNA complexes is indicated in Table 1. Briefly, the transfection reagent (1 μg/μL) and siRNA (0.14 μg/μL), at a 6:1 ratio of transfection reagent/siRNA with a total concentration in the range of 20–60 nM, were added to RPMI 1640 medium and incubated for 30 min to facilitate interaction and form the optimal complexation. For cell treatments, the MOLM-13 or MV4-11 cells (300 μL) were added in each well (48-well plates, ThermoFisher Scientific, Lafayette, CO, USA), and then 100 μL/well of the polyplexes was added. All treatments were performed in triplicate. Complexes of C-siRNA/transfection reagent were used as a negative control, whereas the non-treatment groups were added with only serum-free medium, without the addition of siRNA complexes or treatments. After treatment for 3 and 6 days, cells were collected and analyzed for FLT3 and KMT2A::MLLT3 silencing efficiency.

### 2.4. siRNA Combination Therapy

The FLT3 and KMT2A::MLLT3 siRNAs were delivered to MOLM-13 cells at a 6:1 ratio of polymer/siRNA with a total concentration of 60 nM (30 nM each), utilizing combinational siRNA delivery. Additionally, individual siRNAs targeting FLT3 and KMT2A::MLLT3, each at 60 nM, were delivered using a polymer/siRNA ratio of 6:1 via Prime-Fect and PEI1.2k-PHPA-Lin9 lipopolymers. An MTT assay was utilized to assess the impact of combinational siRNA therapy on cell growth inhibition after 72 h of treatment, as described in Section 2.10. The negative control was represented by treatment with C-siRNA complexes. The utilized polymer/siRNA ratio and the siRNA concentration emerged from preliminary studies and experience, where the focus was to identify the minimal polymer/siRNA ratio as well as the concentration to minimize non-specific effects while keeping the efficacy of the siRNA treatment.

### 2.5. Characterization of Complexes

The formulation of the complexes was prepared in ddH2O at a 6:1 ratio of polymer/C-siRNA (60 nM), incubated for 30 min, and then diluted to a final volume of 1 mL with ddH_2_O. Then, *ζ*-potential and hydrodynamic size were measured using Litesizer 500 (Anton-Paar, Graz, Austria). The morphology of the complexes was characterized by using a JEM 2100 microscope, JEOL at 200 kV to provide detailed imaging. Briefly, a droplet of the complexes was placed on a carbon-coated copper grid and left for 5 min, after which excess liquid was blotted away with filter paper. The sample was stained with 2% uranyl acetate solution for 30 s to achieve negative staining.

### 2.6. SYBR^®^ Green Assay for siRNA Binding

The binding interactions between transfection reagents and siRNA were assessed by using SYBR^®^ Green II staining, which enabled evaluation of their binding efficiency. The complexes were prepared in 96-well plates at different ratios of polymer/siRNA (0.05, 0.1, 0.25, 0.5, 0.75, 1.0, 2.5, 5, 10, and 15 *w*/*w*) and incubated for 15 min at room temperature. SYBR Green solution (1×) in an amount of 200 μL was dispensed into each well, and fluorescence intensity was measured at $\lambda_{ex}$/$\lambda_{em}$ = 485/527 using a multiwell plate reader. The fluorescence of the free C-siRNA alone was set as a reference at 100%, and the binding at different polymer/siRNA ratios was determined based on the reduction in measured fluorescence. The binding capacity was determined based on the 50% binding ratio (BC_50_), observed by plotting binding capacity (%) against polymer/siRNA ratios (*w*/*w*).

### 2.7. siRNA Release by Heparin Competition Assay

The dissociation of the complexes was evaluated by a heparin competition assay. A total of 50 μL of the complexes (polymer/siRNA ratios at 2.5, 5, and 10, *w*/*w*) was mixed with 50 μL of heparin solution (0, 0.25, 0.5, 1, 2.5, 5, 10, and 15 USP unit/mL) and incubated for 30 min. Then, 100 μL of 1× SYBR Green I was added to each well to measure free siRNA. All samples were evaluated in triplicate (*n* = 3). The heparin solutions were added to the appropriate buffer and used as a blank for corresponding samples to obtain more accurate results.

### 2.8. siRNA Uptake in MOLM-13 Cells

FAM-labeled siRNA was employed to monitor siRNA delivery within cells and to evaluate the delivery efficiency of Prime-Fect and PEI1.2k-PHPA-Lin9. Transfection of MOLM-13 cells was performed using FAM-labeled siRNA at 60 nM and a polymer/siRNA ratio of 6:1. Briefly, for preparation of complexes, 2.3 μL of polymer (1.0 μg/μL) was mixed with 2.4 μL of FAM-labeled siRNA (0.14 μg/μL) in RPMI 1640 medium at a final volume of 100 μL. All samples were viewed, and images were taken under an epifluorescent microscope after 24 h of incubation.

### 2.9. Quantitative Analysis of siRNA Uptake in MOLM-13 Cells

Flow cytometry was used to perform a quantitative analysis of FAM-labeled siRNA uptake in the complexes, evaluating the efficiency of cellular delivery. MOLM-13 cells were initially cultured at 100,000 cells/mL in 300 μL per well. Following seeding, they were transfected with 100 μL of FAM-labeled siRNA at 60 nM and a 6:1 ratio of polymer/siRNA. The negative control employed was non-labeled C-siRNA. Cells were collected, washed twice with HBSS buffer after 24 h of treatment, and then fixed with a 3.7% formalin solution. A BD LSR Fortessa-SORP flow cytometer (BD, Biosciences, Frankin Lakes, NJ, USA) was used to quantify the positive population and mean fluorescence of FAM-labeled siRNA-positive cells and a 1% threshold was set to define the FAM-labeled siRNA-positive population.

### 2.10. Cell Growth Inhibition by MTT Assay

The efficacy of FLT3 and KMT2A::MLLT3 siRNA silencing on cell growth inhibition in MOLM-13 cells was investigated using the MTT assay. MOLM-13 cells were placed in wells at 35,000 cells/mL in 300 μL and transfected with 100 μL of the complexes. At the designated time points, 100 μL of MTT solution was added to each well to achieve a final concentration of 1 mg/mL and incubated for an additional 40 min. Cells were transferred into an Eppendorf tube, and the residual medium was carefully discarded. The formazan crystals, which were violet in color, were solubilized by adding 100 μL of DMSO. Following transfer to a 96-well plate, optical density measurements were taken at 570 nm with a microplate reader (SPECTRAmax^TM^ 250, Molecular Devices Corporation, Sunnyvale, CA, USA). The blank solution consisted of 100 μL of pure DMSO. To calculate the percentage of cell viability, the following formula was applied: 100% × (absorbance of cells treated with the complexes/absorbance of untreated cells). To assess significant differences between the study groups, one-way ANOVA via Prism 8.0 software (GraphPad, San Diego, CA, USA) was employed to assess significant differences between the study groups (*p* < 0.05).

### 2.11. qPCR Analysis for Assessing Silencing Activity of siRNAs

The MOLM-13 cells (5 × 10^5^ cells/well) were transfected with the complexes prepared using FLT3 siRNA, KMT2A::MLLT3 siRNA, and C-siRNAs at 60 nM with a polymer/siRNA ratio of 6:1. All samples were transferred into Eppendorf tubes after 24 and 72 h of treatment, then centrifuged, and total RNA was isolated using TRIzol reagent following the manufacturer’s guidelines. The quantity and purity of the RNA were evaluated by measuring the absorbance ratio at 260 and 280 nm with a GE Nanovue spectrophotometer. The SensiFAST cDNA synthesis kit was used to convert 1 μg of total RNA into cDNA following the manufacturer’s recommendations (Meridian Bioscience, OH, USA). Real-time PCR (RT-PCR) was performed using SYBR Green/ROX master mix (2×) (MAF Center, University of Alberta) on a StepOne real-time PCR system (Applied Biosystem, Foster City, CA, USA). The levels of expression of housekeeping endogenous gene human beta-actin were detected by a specific primer (reverse: 5′-AAT TTA CAC GAA AGC AAT GCT ATC A-3′; forward: 5′-CCA CCC CAC TTC TCT CTA AFF A-3′) that was supplied by IDT and designed by the NCBI Primer-BLAST. For qPCR amplifications, a total volume of 10 μL per sample was prepared, containing 3 μL of cDNA template, 5 μL of 2× SYBR Green master mix, and 0.4 μL each of the reverse primer and forward primer (10 μM). The reactions were performed in triplicate samples using a Fast Optical 96-well plate. Negative controls were prepared by omitting the cDNA template from the qPCR reactions. The conditions for qPCR consisted of pre-incubation at 95 °C for 5 min, followed by 40 cycles of denaturation of 95 °C for 15 s, and annealing/elongation of 65 °C for 1 min. The 2^−ΔΔCT^ method was employed to quantify gene expression levels, with target gene cycle threshold (C_T_) values normalized to those of beta-actin, which served as a reference. In addition, the relative quantity of transcripts was reported, and the non-treatment group was used as the calibrator.

### 2.12. Cell Surface FLT3 Quantitation

After transfecting MOLM-13 cells with the complexes, the cells were transferred into Eppendorf tube at the desired times, and centrifuged, and the supernatant was removed. The cells were washed twice with HBSS and then resuspended in 300 uL of HBSS with 10% *v*/*v* of FBS. For staining, Alexa Fluor^®^ 647 mouse IgG1κ isotype control (Clone MOPC-21) or anti-human FLT3 monoclonal antibody conjugated with Alexa Fluor^®^ 647 (Clone 4G8) was added at a 1:1 dilution (1 µL), and staining was performed in a dark environment for 40 min at 4 °C. Excess antibody was removed by washing twice with HBSS. Subsequently, 200 µL of 1.7–1.8% formaldehyde was added; the fixed cells were then transferred to flow cytometry tubes and measured with a BD LSR Fortessa-SORP flow cytometer (BD Biosciences, Frankin Lakes, NJ, USA).

### 2.13. Colony-Forming Cell (CFC) Assay

Briefly, the complexes at a 6:1 ratio of polymer/siRNA were used to treat MOLM-13 cells at 60 nM for the inhibition of cell proliferation. After incubating for 24 h, cells were counted using a hemocytometer and the trypan blue stain for viable cell numbers. A total of 4000 cells were transferred to methylcellulose media and mixed. Then, the samples were transferred into 24-well plates in the center wells (excluding wells at the borders), and the colonies were counted using optic microscopy after 2 weeks of incubation.

### 2.14. Analysis of Apoptosis

After 24 h of treatment with complexes at 60 nM and a polymer/siRNA ratio of 6, cells were collected and washed twice with the PBS. Then, 1× Annexin V binding buffer (100 μL) was added, and all samples were transferred to polystyrene round-bottom tubes. Then, using an Apoptosis Kit from BD Biosciences, 2.5 μL each of FITC-Annexin V and Propidium Iodine were added and incubated for 15 min. The flow cytometry histograms were analyzed to obtain the FITC-Annexin-positive cell population (designated as the early apoptosis population) and PI-positive cell population (designated as the late apoptosis population). The data were subsequently normalized with non-treated cells (NT; designated as 1% in both cell populations) to obtain the relative ratio of each population.

### 2.15. Animal Study

Animal experiments were performed following procedures pre-approved by the Health Sciences Laboratory Animal Services (HSLAS), University of Alberta (Ethics Approval AUP00000423, 25 July 2024). We purchased male, triple immunodeficient NOD-*Prkdc^em26Cd52^Il2rg^em26Cd22^* (NCG) mice, aged 6–8 weeks, from Charles River Laboratories (Laval, QB, Canada). MOLM-13 Luc+ cells (3 × 10^5^) in 100 µL of RPMI medium were injected intravenously through the tail vein to induce leukemia engraftment. The mice were randomly assigned to treatment groups (*n* = 5 in each group) and received intraperitoneal injections of lipopolymer/siRNA complexes (25 µg siRNA per mouse, lipopolymer/siRNA ratio of 7.5:1) on days 02, 04, 06, 08, 10, 12, and 14. Leukemia progression was monitored using the IVIS^®^ Spectrum Imaging System (PerkinElmer, Inc., Waltham, MA, USA) on days 01, 05, 09, and 13. Briefly, mice were injected intraperitoneally with D-Luciferin (Morton Grove, IL, USA) at a concentration of 3.9 mg/mouse. After 10 min, the mice were placed on a prewarmed stage inside an IVIS^®^ light-tight chamber equipped with an XGI-8 anesthesia system for imaging. Image analysis was performed using Living Image^®^ 4.8.0 software (PerkinElmer, Inc., Waltham, MA, USA). Mice were humanely euthanized on day 15. Changes in luminescence values were plotted for individual mice and normalized with day 5 values in order to account for different growth rates of the grafts observed among the mice in each group.

### 2.16. Statistical Analysis

The data are expressed as mean ±/+ standard deviation (SD). Statistical analysis was performed using one-way ANOVA followed by Tukey’s test in Prism8 (GraphPad software, San Diego, USA) to compare scrambled siRNA groups and specific siRNA groups. Significance levels are denoted as follows: *: *p* ≤ 0.05; **: *p* ≤ 0.01; ***: *p* ≤ 0.001; ****: *p* ≤ 0.0001.

## 3. Results

### 3.1. Characterization of Polymer/siRNA Complexes

To investigate whether complexes possess the expected nanoparticle characteristics for efficient siRNA delivery to MOLM-13 cells, we examined siRNA binding to polymers, complex stability, *ζ*-potential (surface charge), and the hydrodynamic size of complexes. The binding capacity of siRNA for Prime-Fect and PEI1.2k-PHPA-Lin9 at different ratios of polymer/siRNA was determined using the SYBR Green assay (Figure 1a). The intercalating nucleic acid dye SYBR Green binds to free and unbound siRNA, leading to the induction of a fluorescence signal. Figure 1a summarizes the highly efficient siRNA binding of Prime-Fect and PEI1.2k-PHPA-Lin9, showing that both polymers fully bound to siRNA at ratio of polymer/siRNA below 2.0. Prime-Fect and PEI1.2k-PHPA-Lin9 displayed a BC_50_ value at polymer/siRNA ratios of 0.22 and 0.14, respectively. Binding siRNA at low polymer concentrations allows for minimal inclusion of lipopolymers in the siRNA formulations, avoids cellular exposure to polymers, and minimizes unwanted toxic effects.

The summary of the dissociation of complexes is shown in Figure 1b and indicates that the release of siRNA increased with the higher heparin concentrations incubated. For Prime-Fect/siRNA complexes, a half-maximal dissociation concentration (taken as DC_50_) was observed at 0.75, 2.15, and 4.50 U/mL of heparin, whereas the DC_50_ values for PEI1.2k-PHPA-Lin9 complexes were 0.45, 0.82, and 2.12 U/mL of heparin at polymer/siRNA ratios of 2.5, 5, and 10, respectively. Nevertheless, under the buffer conditions employed in the assay, complete release was not achieved. Accordingly, an increase in the polymer/siRNA ratios enhanced the binding between the lipopolymers and the siRNA (i.e., reduced propensity for dissociation of complexes), and the polymer PEI1.2k-PHPA-Lin9 gave more readily dissociating complexes under the experimental conditions.

The average hydrodynamic size and *ζ*-potential were measured at a polymer/siRNA ratio of 6:1. Prime-Fect/siRNA and PEI1.2k-PHPA-Lin9/siRNA complexes demonstrated average sizes of 210 and 220 nm, respectively, as shown in Figure 1c. These particle sizes are considered adequate for effective cellular internalization and are consistent with prior published data for Prime-Fect/siRNA complexes [37]. The Prime-Fect/siRNA polyplexes showed slightly higher positive charges compared to PEI1.2k-PHPA-Lin9/siRNA polyplexes with the *ζ*-potentials of 25.7 mV and 17.0 mV, respectively.

The TEM images in Figure 1d depict siRNA complexes with Prime-Fect and PEI1.2k-PHPA-Lin9 at a polymer/siRNA ratio of 6:1. Prime-Fect/siRNA complexes had a spherical shape, and the agglomerates were seen in some cases in a close-packed conformation (lower right portion of Figure 1d(i)). The agglomerated particles were most likely the outcome of the TEM processing, particularly the drying procedure [38]. The PEI1.2k-PHPA-Lin9/siRNA complexes displayed uniform particle sizes, characterized by individual spherical particles with smooth borders and no evidence of agglomeration (Figure 1d(ii)).

### 3.2. Uptake of siRNA in MOLM-13 Cells

The intracellular uptake of siRNA into MOLM-13 cells was assessed and quantified using fluorescence microscopy and flow cytometry, respectively (Figure 2). The incubation of MOLM-13 cells with FAM-labeled siRNA in a 6:1 polymer/siRNA ratio and at 60 nM was carried out for 24 h (Figure 2a); punctuate, green FAM-fluorescence could be observed in the cytoplasm of cells, suggesting that the polyplexes had been internalized by the cells. Qualitative results of this low-resolution microscopy analysis indicated higher intracellular siRNA delivery with Prime-Fect in comparison to PEI1.2k-PHPA-Lin9. Detailed analysis of the intracellular trafficking of the complexes with confocal microscopy was not conducted, but quantification of intracellular uptake after treatment with FAM-labeled siRNA was performed using flow cytometry, based on the positive cell population (%) and mean fluorescence intensity/cell (Figure 2b). When compared to the background, both polyplexes displayed significantly higher uptake (*p* ≤ 0.05). A higher siRNA uptake was obtained with the Prime-Fect complexes with the percentage of the FAM-siRNA-positive population being >90%, which was higher than that of the PEI1.2k-PHPA-Lin9 complexes (~55%).

### 3.3. Inhibition of Cell Growth Through FLT3 Silencing

The functional efficacy of polymer/siRNA complexes was determined by delivering FLT3-targeting siRNA (siFLT3) in MOLM-13 cells. The MTT assays were conducted to determine cell growth after 72 h of treatment at 60 nM complex formulations at polymer/siRNA ratios of 2:1, 4:1, and 6:1. The Prime-Fect/siFLT3 and PEI1.2k-PHPA-Lin9/siFLT3 complexes could only demonstrate a slight reduction in cell growth at a low polymer/siRNA ratio of 2:1, with decreases in cell growth of 15.1% and 20.6%, respectively (Figure 3a). At a 4:1 ratio of polymer/siRNA, the Prime-Fect/siFLT3 complexes achieved a 53.0% reduction in cell growth, whereas the PEI1.2k-PHPA-Lin9/siFLT3 complexes achieved a 65.9% decrease. Increasing the ratio of polymer/siRNA to 6:1 achieved the strongest growth inhibition, resulting in 78.7% and 86.0% inhibitions for Prime-Fect/siFLT3 complexes and PEI1.2k-PHPA-Lin9/siFLT3 complexes, respectively (Figure 3a). Prime-Fect-treated cells also gave longer inhibition of cell growth up to day 6 (Appendix A).

The influence of siRNA treatments on target mRNA levels was investigated by treating MOLM-13 cells with C-siRNA and siFLT3 at 60 nM and measuring the mRNA levels by qPCR after 1 and 3 days oftreatment (Figure 3b). With Prime-Fect/siFLT3 complexes, targeted mRNA levels after a single treatment were ~43% on day 1 and ~36% lower after 3 days of treatment (Figure 3b(i)), as compared to the control treatment group of C-siRNA complexes. The PEI1.2k-PHPA-Lin9/siFLT3 complexes displayed ~32% and ~58% downregulation of FLT3 on day 1 and day 3, respectively (Figure 3b(ii)).

Quantification of cell surface FLT3 protein levels by immunochemistry/flow cytometry revealed ~49% and ~38% reductions in the percentage of the FLT3(+) population at 24 h for Prime-Fect and PEI1.2k-PHPA-Lin9, respectively (Figure 3c(i)). The percentage of the FLT3(+) population displayed no significant difference for the control vs. FLT3-specific siRNA delivered with Prime-Fect at 72 h (~47%), while it was slightly decreased to ~28% with PEI1.2k-PHPA-Lin9 (Figure 3c(iii)), thereby confirming the active gene silencing seen in Figure 3b. Additionally, Figure 3c(ii) and Figure 3c(iv) display the mean fluorescence intensities (a.u.) for 24 h and 72 h of treatment, respectively. The reduction in mean fluorescence is also consistent with the result of silencing based on the percentage of the FLT3(+) population.

The functional efficacy of polymer/siRNA complexes was also determined by delivering FLT3 siRNA in another AML cell line, MV4-11 cells. The MTT assays were conducted for cell growth after 72 h of treatment with 60 nM siRNA and with complex formulations with polymer/siRNA ratios of 5:1 and 7.5:1. The Prime-Fect/siFLT3 and PEI1.2k-PHPA-Lin9/siFLT3 complexes demonstrated significant reductions in cell growth at both ratios (Figure 4). On the average, ~40% reduction in cell growth was seen for the specific siFLT3 treatment under various conditions.

### 3.4. Combinational Silencing of FLT3 and KMT2A::MLLT in MOLM-13 Cells

To study the ability of polymers to transport dual siRNAs simultaneously targeting FLT3 and KMT2A::MLLT3 and their effect on leukemic cell growth, MOLM-13 cells were treated with the corresponding complexes. The treatment was carried out in three groups: (i) complexes with single target-specific siRNA, (ii) complexes containing a 1:1 ratio of C-siRNA and a target-specific siRNA, and (iii) complexes containing two target-specific siRNAs at a 1:1 ratio. The final concentrations of siRNA were 40, 60, and 80 nM in tissue culture medium. The results (Figure 5) indicated that both polymers effectively delivered the siRNAs and influenced the MOLM-13 cell growth, in contrast to treatment with polymer/C-siRNA complexes. As the siRNA concentration increased, a similar trend of decreasing growth was seen with both polymers. The results with the combination of siRNAs (i.e., the FLT3 + KMT2A::MLLT3 siRNA combination) were most potent with the complexes of both polymers, leading to cell growth of only 20% or less. The statistical variations for the treatment groups are summarized in Table 2; based on this analysis, Prime-Fect complexes appeared to be more effective among the study groups as compared to the PEI1.2k-PHPA-Lin9 complexes.

### 3.5. Assessment of Cell Growth Inhibition Using the CFC Assay

The impact of siRNA treatment on MOLM-13 cell proliferation was further investigated by the CFC assay, which involved longer-term incubation of cells upon treatment. In this experiment, the cells were transfected with the complexes for 24 h before being transferred to methylcellulose media for a 14-day incubation period. There were fewer colonies with siFLT3- and siKMT2A::MLLT3-treated MOLM-13 cells when compared with the untreated control (Figure 6a). With single siRNA complexes, siFLT3 showed a significant decrease (*p* ≤ 0.05) in colony counts in comparison with C-siRNA groups for Prime-Fect (27.0 ± 0.3% vs. 78.7 ± 2.8%) and PEI1.2k-PHPA-Lin9 (31.0 ± 2.4% vs. 84.3 ± 13.8%) (Figure 6b). The siKMT2A::MLLT3 treatment gave 39.6 ± 5.8% and 49.3 ± 5.3% with Prime-Fect and PEI1.2k-PHPA-Lin9, respectively (*p* ≤0.05; Figure 6b). No significant reduction in colony counts was observed with the combinations of C-siRNA+siFLT3 (30 + 30 nM) or C-siRNA+siKMT2A::MLLT3 (30 + 30 nM) when compared to C-siRNA treatment. However, the combination of siFLT3 and siKMT2A::MLLT3 (30 + 30 nM) displayed a significant decrease in colony counts compared to C-siRNA, C-siRNA+siFLT3, and C-siRNA+ siKMT2A::MLLT3 treatments: 27.7 ± 4.2% and 32.2 ± 4.1% with Prime-Fect and PEI1.2k-PHPA-Lin9, respectively (*p* ≤ 0.05; Figure 6b).

### 3.6. Apoptosis Analysis

The Annexin-V/PI assay was performed and analyzed via flow cytometry to evaluate cellular apoptosis. In this assay, cells in early apoptosis are marked by positive Annexin V and negative PI staining, whereas cells in late apoptosis show positive staining for both Annexin V and PI. The early apoptotic population was elevated in the Prime-Fect with single siRNA treatment groups (siFLT3 or siKMT2A::MLLT3) and the combination treatment, as compared to the C-siRNA-treated group (Figure 7a). Unlike Prime-Fect, cells treated with PEI1.2k-PHPA-Lin9/siRNA complexes exhibited no significant early apoptotic population (Figure 7a). The combination group (siFLT3+siKMT2A::MLLT3) with Prime-Fect also gave significantly higher early and late apoptotic populations than the C-siRNA, C-siRNA+siFLT3, and C-siRNA+ siKMT2A::MLLT3 groups (Figure 7a,b and Appendix A). With PEI1.2k-PHPA-Lin9/siRNA complexes, the late apoptotic populations in the single KMT2A::MLLT3 siRNA treatment and combination (FLT3+KMT2A::MLLT3 siRNAs) were higher than in the C-siRNA treatment (Figure 7b).

### 3.7. Animal Study

To explore whether the siRNA complexes were also effective in an animal model, Luc+ MOLM-13 cells were grafted in NCG mice by intravenous (tail vein) injection, and in vivo bioluminescence imaging on the mice was employed to evaluate the progression of leukemia. We then compared the growth of Luc+ MOLM-13 cells in vivo after treatment with C-siRNA and FLT3 siRNA complexes formulated with Prime-Fect and PEI1.2-PHPA-Lin9. The in vitro response of the Luc+ MOLM-13 cells to treatment with siRNA complexes is shown in Appendix A. An attenuated response with Luc+ MOLM-13 cells was evident compared to the native MOLM-13 cells used in previous experiments; only FLT3 siRNA delivered with Prime-Fect was effective in these cells, while delivery with PEI1.2-PHPA-Lin9 was not effective. Nevertheless, we used both polymers to treat the xenografted Luc+ MOLM-13 cells, considering that in vivo conditions could be significantly different from cell culture conditions.

The treatment regimen with siRNA complexes (25 μg siRNA per injection, corresponding to ~1 mg/kg) involved IP injection every 2 days starting on the day of engraftment, as indicated in Figure 8a. Bioluminescent imaging was performed on days 5, 9, and 13, where the data from day 5 were used to normalize the tumor burden in each mouse (to account for initial variations in tumor burden). Based on this analysis, treatments with the Prime-Fect/siFLT3 complex significantly inhibited the leukemia burden by day 13, in contrast to the Prime-Fect/C-siRNA group (Figure 8b,c). The PEI1.2-PHPA-Lin9 system did not achieve a reduction in leukemia burden (Figure 8d,e), in line with the effects seen on Luc+ MOLM-13 cells in vitro (Appendix A). Due to the aggressive nature of these leukemia cells, humane euthanasia was required on day 15.

## 4. Discussion

Leukemias, characterized by abnormal proliferation of sub-populations of hematopoietic stem cells, is typically driven by mutated genes that impart oncogenic features to host cells. The mutational landscape of various leukemias is well studied and represents a clear setting for development of specific therapies. In this regard, the oncogenes that induce a ‘gain-of-function’ are suitable targets for RNAi and its pharmacological mediator siRNA. Some over-activated oncogenic drivers are broadly distributed in host cells, such as STAT5A [39], which requires their silencing specifically in leukemic cells without compromising the expression in the hematopoietic compartment. Other oncogenic drivers are more specific, such as the example of fusion oncogenes Bcr-Abl. In this study, we targeted two oncogenes, one that is broadly expressed—FLT3 tyrosine kinase—and one highly restricted—KMT2A::MLLT3. In the former case, our siRNA was designed to target the native FLT3 and not the internal tandem duplicated FLT3 (FLT3-ITD), which is also highly concentrated in leukemic cells. We show that both oncogenes are viable targets for the siRNA delivered by non-viral means in this study. Our previous targets additionally explored the Bcr-Abl oncogene and the transcription factor STAT5A in control of leukemic cell growth using similar non-viral techniques, so that the proposed delivery system could be a fruitful platform in RNAi-mediated leukemia therapy in general. Moreover, our studies utilized a range of leukemic cells, such as Bcr-Abl(+) cells, FLT3-ITD(+) MV4;11 cells, STAT5-A over-expressing RS4:11 cells, and primary mononuclear cells obtained from leukemia patients (bone marrow isolates), so that the RNAi platform is adoptable to a range of well-accepted leukemia cell models.

We focused on two non-viral carriers in this study, based on preliminary studies where the carriers were screened for efficacy of siRNA delivery. There was no major discernable difference in the interactions of the lipopolymers with the siRNA and resultant size/charge of complexes; an obvious difference was the reduced propensity of Prime-Fect to dissociate in the presence of the competing polyanion heparin. It is possible that this difference also resulted in the increased delivery of siRNA to MOLM-13 cells, based on quantitation of FAM-siRNA in cells by flow cytometry.

The silencing efficiency, however, was noted to be similar in MOLM-13 cells, where Prime-Fect provided a quicker silencing at the mRNA level (day 3), but PEI1.2k-PHPA-Lin9 a longer-lasting silencing (day 6). The reductions in FLT3 protein levels were similar between the two delivery systems, based on mean levels of protein detected by immunochemistry staining. The long-term functional outcome evaluated upon siRNA-mediated silencing was based on colony formation in agarose gels; in this assay, both lipopolymers gave a similar suppression of colony formation, when one considered the delivery of FLT3 siRNA alone, KMT2A::MLL3 siRNA alone, or their combination (data from Figure 7). A significant difference was seen in early apoptotic markers as assayed by AnnexinV/PI staining; Prime-Fect-delivered siRNAs were more effective in inducing early apoptosis (AnnexinV-positive cells), while cells in the late apoptotic phase (AnnexinV/PI-positive cells) were generally equivalent between the two carriers. Of note was the generally lower levels of apoptotic cells with PEI1.2k-PHPA-Lin9 treatment, which may be indicative of lower interaction of this polymer with the cells and/or its more biocompatible nature when used for delivery of non-specific (control) siRNA.

We found both of the explored targets (FLT3 and KMT2A::MLLT3) therapeutically effective when it came to controlling unwanted leukemic cell growth. It was our thought that we could enhance the efficacy of our siRNA therapy when we targeted both mediators with specific siRNAs, based on previous studies that showed the activated FLT3 tyrosine kinase receptor can act in tandem with KMT2A::MLLT3 to accelerate the onset of leukemogenesis [40]. In both the growth inhibition assay as well as colony formation and apoptosis assays, there was no evidence that the combination of both targets led to synergistic activities. Usually, the combination of FLT3 and KMT2A::MLLT3 siRNAs (30 + 30 nM) was equally as effective as FLT3 siRNA alone (60 nM), more so than the KMT2A::MLLT3 siRNA alone. It is possible that the relatively homogeneous MOLM-13 cells did not display variations that allowed synergistic activities and that primary patient samples could display more heterogeneity that can manifest the desired synergistic activity. We also kept the total dose of siRNA equivalent in this comparison (60 nM in total), and did not want to enhance the dose used in the culture in order to better compare the outcomes; it is possible that if the dose of FLT3 siRNA is maintained constant (60 nM) and KMT2A::MLLT3 siRNA is simply added on top of this dose, more potent or synergistic activity could be observed. The non-specific toxicity manifested with the complexes could have also been greater in that case, so we wanted to avoid this situation. Further studies will be required in this regard to better identify synergistic pairs of targets for silencing. We think that this is an important consideration to ultimately lower the total dose of siRNA needed, as well as better control leukemic growth.

Ultimately, preclinical evaluation of the siRNA approach has led to therapeutic outcomes in an animal model for clinical translation. Unlike our previous studies [41,42], this study relied on Luc(+) cells to quantitate the total leukemic burden in animals. FLT3 was chosen as the target in these studies since the various in vitro assays indicated its silencing to be more potent in inhibiting cell growth than inhibiting the fusion protein KMT2A::MLLT3. We observed an attenuated response to our siRNA therapy with Luc(+) cells in vitro as compared to the native phenotype; the response to FLT3 siRNA was lower with Prime-Fect and minimal, if any, with PEI1.2k-PHPA-Lin9 complexes. Nevertheless, we proceeded with animal studies to demonstrate the principle of siRNA therapy on systemic tumors. In parallel with in vitro results, results in the animal model also indicated the Prime-Fect complexes of FLT3 siRNA to be effective in reducing the systemic tumor burden, unlike the PEI1.2k-PHPA-Lin9 complexes. The effect was evident after day 13, possibly indicating the need for the full duration of siRNA treatment to observe the desired therapeutic effect. Despite its significance, the inhibition of growth was relatively small, unlike in our other studies which investigated the growth of local tumors [41], where >50% inhibition of growth was evident within the study periods. We plan to undertake future studies by using complex formulations specifically designed for Luc(+) cells so that the factors affecting in vivo therapeutic outcome can be better elucidated.

The combination of FLT3 and KMT2A::MLLT3 siRNAs enhanced the effect of delivering either FLT3 or KMT2A::MLLT3 siRNA alone. The combination siRNA treatment resulted in cell growth inhibition, reducing cell viability by ~29% and ~37% for Prime-Fect and PEI1.2k-PHPA-Lin9, respectively, at a polymer/siRNA ratio of 6:1 after three days of treatment. While these studies focused primarily on one cell type (MOLM-13), the response to siRNA therapy (with FLT3 siRNA alone) was similar in another AML model, MV4;11 cells (Figure 4). More in-depth studies with the MV4;11 cells were reported separately [41], which confirmed the potential of the described lipopolymer complexes in another cell model of AML. Hence, collectively, the present study demonstrates the potential of lipopolymers in achieving efficient siRNA therapy and provides evidence for the feasibility of exploring different siRNA combinations targeting different oncogenes, which is relevant in leukemia cancer. We note, however, that our studies to date were conducted with male mouse models only, and future studies should test whether the response to the siRNA therapy is dependent on the sex of the chosen animal model.

While promising, the non-viral siRNA therapy is also associated with certain drawbacks, some of which are articulated in Dezfouli et al. [43]; these include (i) non-specific actions of siRNA due to cross-reactivity with mRNAs with similar sequences to the target mRNA; (ii) non-specific cytotoxicity associated with delivery systems, which are often excessively cationic and fusogenic; and (iii) over-loading of the RNAi machinery by the exogenous siRNA at the expense of native regulatory RNAs. These shortcomings could be mitigated by deploying potent delivery systems that function at relatively low doses (both in siRNA and delivery material amounts), administered via more acceptable routes. Delivery via electrostatically interacting siRNA nanoparticles is relatively convenient to prepare and administer, as long as pre-mature release of the siRNA cargo is minimized (to prevent degradation of the siRNA) and complete release of the siRNA is facilitated at the site of action (to make the active pharmacological agent fully available to RNAi machinery). These convenient nanoparticles can protect the siRNA from degradation, the primary limitation for using naked siRNA as a pharmacological agent, based on confirming the stability of siRNA in serum-containing media in vitro (using electrophoretic techniques) [41] and the recovery of intact siRNA in biodistribution studies in animal models [44]. Otherwise, relatively high doses need to be applied with siRNA therapy, leading to unwanted effects.

## 5. Conclusions

In this study, we present a siRNA delivery system, derived from lipopolymers PEI1.2k-PHPA-Lin9 and Prime-Fect, that is suitable for treatment of the AML cell line MOLM-13. The delivery systems explored here are distinctly different from conventional delivery systems that are based on LNPs. The latter consist of a mixture of lipids (usually three or four), while our delivery system is a single molecular component constructed by grafting a lipid onto a small cationic polymer. Effective delivery of siRNA to leukemia cells was shown with this class of delivery systems, unlike the LNPs [41]. A new leukemia model (MOLM-13 cells) was employed for testing the lipopolymers in culture and in a xenograft model, leading to original observations and further confirming the feasibility of lipopolymers for treatment of leukemia cancers in experimental models. The PEI1.2k-PHPA-Lin9 and Prime-Fect complexes, with surface charges at +17 mV and +25 mV, respectively, and hydrodynamic diameters around 200–220 nm, displayed relatively high dissociation efficacy. At weight ratios of polymer/siRNA below 2, both polymers were capable of completely binding to siRNA, resulting in spherical particles with no signs of agglomeration. The Prime-Fect complexes achieved a higher siRNA delivery efficiency, with a positive population percentage of over 90%, which was higher than that of the PEI1.2k-PHPA-Lin9 complexes. Both FLT3 and KMT2A::MLLT3 were viable targets to reduce growth of the AML cells after siRNA treatment under a variety of settings, and the combination of FLT3 and KMT2A::MLLT3 siRNAs enhanced the effect of delivering individual siRNAs, although full synergistic activity was not seen. In line with the cell culture results, the FLT3 siRNA complexes formed with Prime-Fect were more effective in reducing the growth of MOLM-13 xenografts in the NCG mouse model. We believe that the superb precision of the siRNA approach among the current pharmacological interventions is highly promising, so that implementing this mode of treatment in clinical leukemia therapy will yield significant benefits over the current treatments.

## Figures and Tables

**Figure 1 biomolecules-15-00115-f001:**
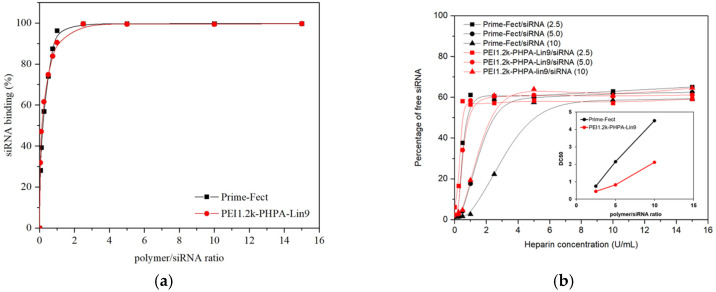
(**a**) siRNA binding profiles of complexes at different ratios of polymer/siRNA were evaluated using the SYBR Green II dye exclusion assay (*n* = 3). (**b**) Dissociation of polymer/siRNA complexes incubated with different concentrations of heparin. The insert displays the variation in DC_50_ values relative to the polymer/siRNA ratio. (**c**) Hydrodynamic size in nm (mean + SD) and *ζ*-potential (mean ± SD) of Prime-Fect and PEI1.2k-PHPA-Lin9 complexes (*n* = 3). (**d**) Morphology of polymer/siRNA complexes at a 6:1 ratio (*w*/*w*) by TEM: (i) Prime-Fect/siRNA and (ii) PEI1.2k-PHPA-Lin9/siRNA.

**Figure 2 biomolecules-15-00115-f002:**
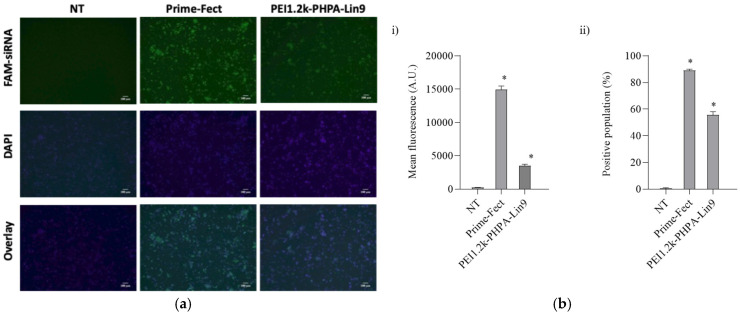
Cellular uptake of Prime-Fect/FAM-siRNA and PEI1.2k-PHPA-Lin9/FAM-siRNA complexes in MOLM-13 cells after transfection with complexes at 60 nM siRNA concentration and using 6:1 polymer/siRNA ratios for 24 h. (**a**) Visualization of FAM-siRNA uptake by epifluorescent fluorescence microscopy. The scale bar represents 100 μm. (**b**) Quantification of intracellular uptake after treatment with FAM-labeled siRNA complexes by flow cytometry. (i) Mean fluorescence intensity of FAM-siRNA per cell. (ii) Percentage of FAM-siRNA-positive cell population. Mean + 1 SD is shown (*n* = 3). * *p* ≤ 0.05 versus C-siRNA.

**Figure 3 biomolecules-15-00115-f003:**
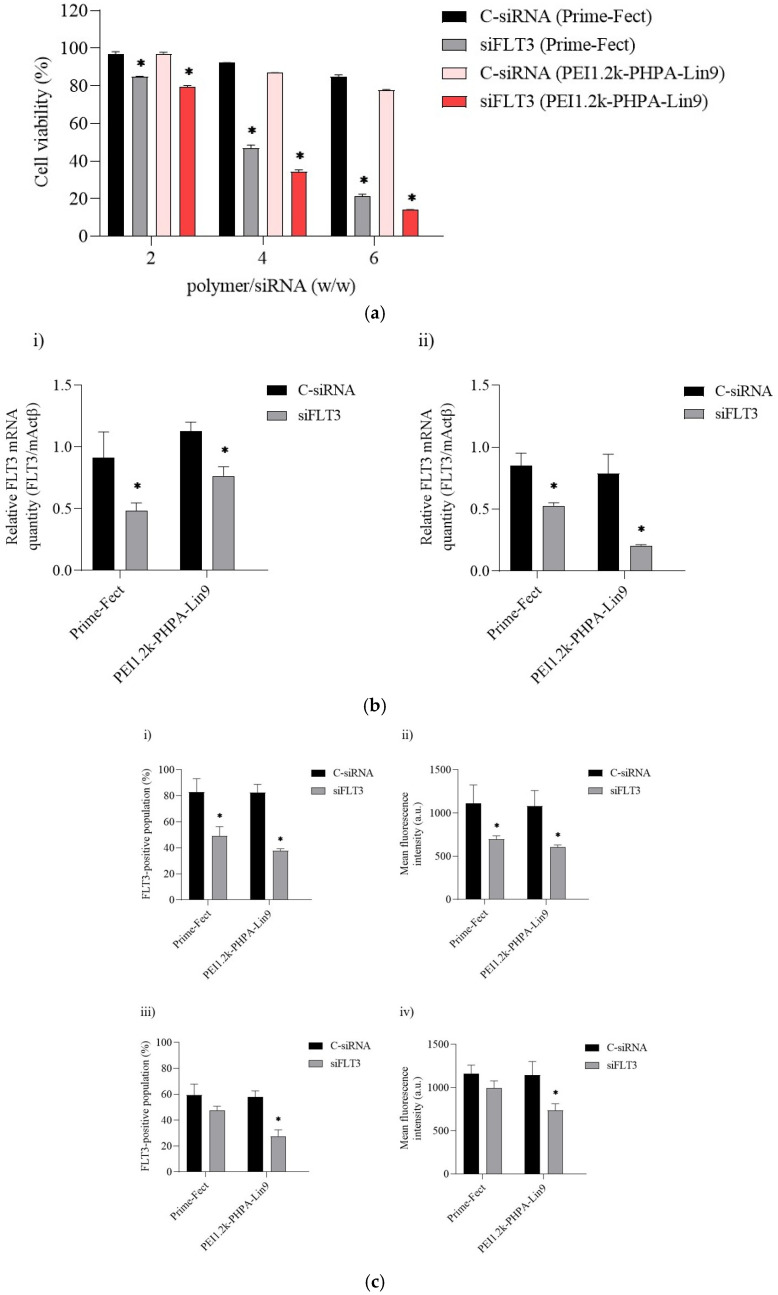
(**a**) Effect of the polymer/siRNA ratio on cell viability. The percentage of cell viability is summarized for MOLM-13 cells treated with single siRNA (C-siRNA and FLT3 siRNA) complexes with Prime-Fect and PEI1.2k-PHPA-Lin9 prepared at different ratios of polymer/siRNA (2, 4, and 6). An MTT assay was conducted to assess cell viability after 72 h of transfection, and it was normalized with the untreated group (i.e., taken as 100% cell viability as a control). * *p* ≤ 0.05 versus C-siRNA. (**b**) FLT3 gene silencing in MOLM-13 cells as evaluated by qPCR. Cells were transfected with Prime-Fect/siRNA and PEI1.2k-PHPA-Lin9/siRNA complexes for 1 day (i) and 3 days (ii) using 60 nM siRNA concentration and a ratio of polymer/siRNA at 6:1. C-siRNA was employed as a control and significance was determined with * *p* ≤ 0.05 in comparison with C-siRNA. (**c**) FLT3 protein levels in MOLM-13 cells were determined by immunochemistry/flow cytometry analysis. (i) FLT-3-positive cell population and (ii) mean FLT3 levels per cell (given by arbitrary fluorescence units of FLT3 antibody) after 24 h of treatment. (iii) FLT-3-positive cell population and (iv) mean FLT3 levels per cell after 72 h of treatment. * *p* ≤ 0.05 versus control siRNA.

**Figure 4 biomolecules-15-00115-f004:**
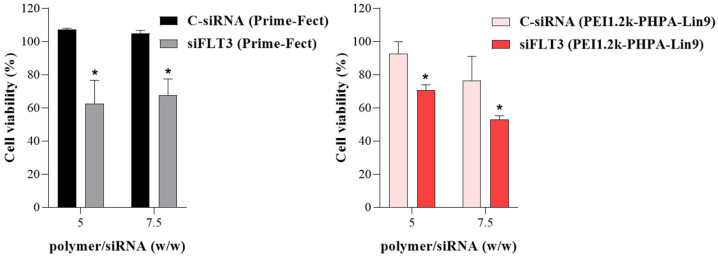
The percentage of cell viability is summarized for MV4-11 cells treated with single siRNA (C-siRNA and FLT3 siRNA) complexes with Prime-Fect (**left**) and PEI1.2k-PHPA-Lin9 (**right**) prepared at polymer/siRNA ratios of 5 and 7.5. The MTT assay was conducted to assess cell viability after 72 h of transfection, and it was normalized with the untreated group (i.e., with 100% cell viability taken as a control). * *p* ≤ 0.05 versus control siRNA.

**Figure 5 biomolecules-15-00115-f005:**
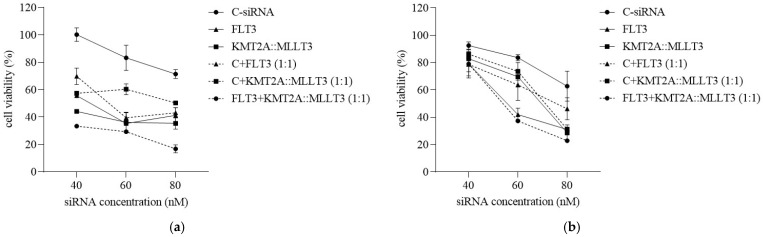
Inhibition of MOLM-13 cell growth with siRNA complexes targeting FLT3 and KMT2A::MLLT3. Percentages of cell viabilities are summarized with treatment of single and combinational siRNA (C-siRNA, siFLT3, and KMT2A::MLLT3) complexes, prepared with (**a**) Prime-Fect, and (**b**) PEI1.2k-PHPA-Lin9 at different siRNA concentrations (40, 60, and 80 nM).

**Figure 6 biomolecules-15-00115-f006:**
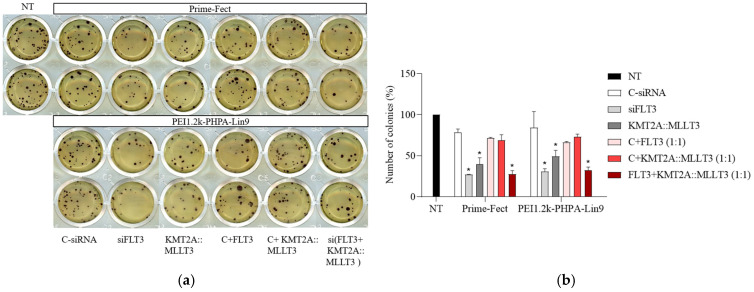
Colony formation in MOLM-13 cells after siRNA treatment. The treatment consisted of polymer complexes with C-siRNA, FLT3 siRNA, KMT2A::MLLT3 siRNA, and an FLT3+ KMT2A::MLLT3 siRNA combination at 60 nM (total siRNA concentration) and a ratio of polymer/siRNA of 6:1. The colonies were counted after two weeks of treatment. (**a**) The representative images of the colony formation after being stained with the MTT solution. (**b**) Quantification of the colony numbers, showing the differences in colony formation reduction compared to the untreated group. Mean + SD (*n* = 3) values are reported, with statistically significant groups noted at * *p* < 0.05 vs. C-siRNA.

**Figure 7 biomolecules-15-00115-f007:**
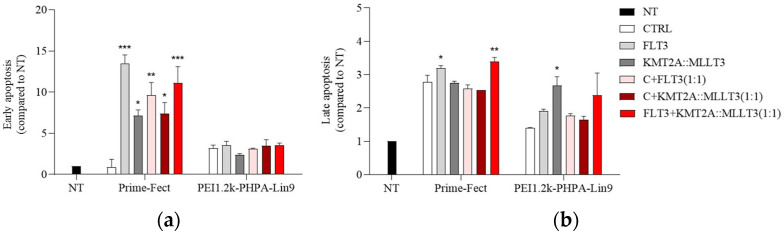
Assessment of early and late apoptosis in MOLM-13 cells using an Annexin-V/PI apoptosis assay after treatment with the complexes. MOLM-13 cells were treated with the complexes targeting FLT3, KMT2A::MLLT3, or their combination at 60 nM. The apoptosis assay was conducted after 72 h of treatment. (**a**) Representative (%) number of cells undergoing early-stage apoptosis. (**b**) Representative (%) number of cells in the late stage of apoptosis. Mean + SD is provided from triplicate experiments with significance indicated by *: *p* ≤ 0.05, **: *p* ≤ 0.01, and ***: *p* ≤ 0.001.

**Figure 8 biomolecules-15-00115-f008:**
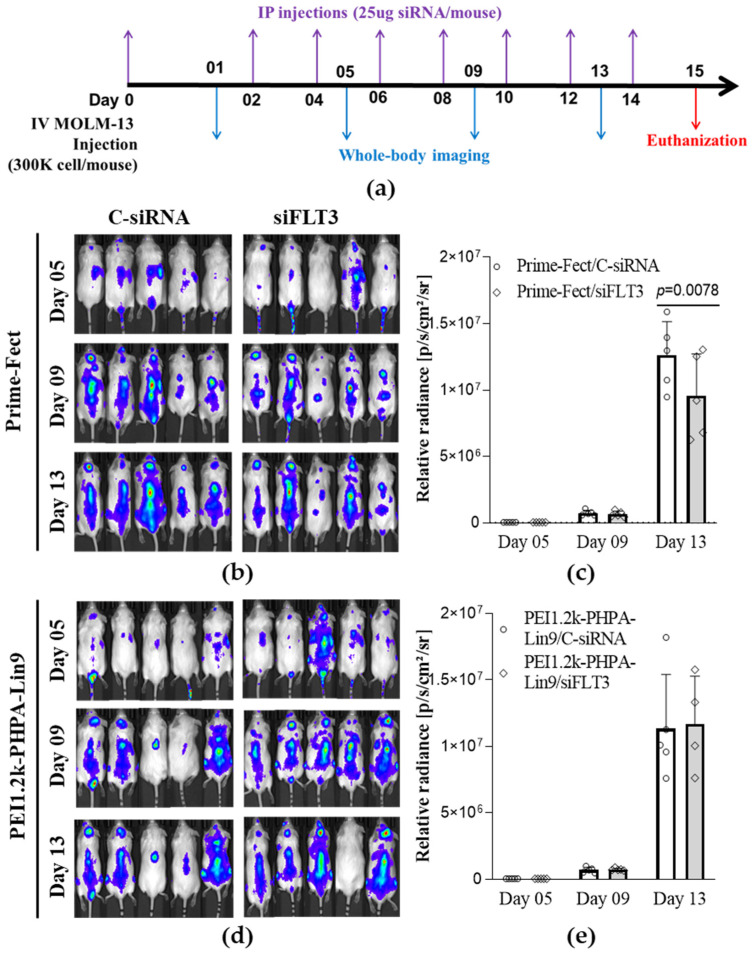
Effects of FLT3 siRNA complexed with Prime-Fect and PEI1.2k-PHPA-Lin9 on Luc+ MOLM-1l xenografts in NCG mouse model. (**a**) Timing scheme for the siRNA treatments and the whole-body imaging. (**b**,**d**) Whole-body bioluminescence images of mice on days 5, 9, and 13, showing leukemia engraftment extent in vivo. (**c**,**e**) Quantification of in vivo bioluminescence to evaluate the reduction in leukemia burden in mice (an error occurred in the whole-body imaging of the fourth mouse in the PEI1.2k-PHPA-Lin9/siFLT3 group on day 13; consequently, this was excluded from the analysis). Data are presented as mean ± SD (*n* = 5). Significant differences from the C-siRNA group were analyzed by multiple *t*-tests (Holm–Sidak test).

**Table 1 biomolecules-15-00115-t001:** The study design for preparing the siRNA complex for a single well. The siRNA concentration indicates the final concentration in the well. The transfection reagent/siRNA ratio is given based on the weight/weight ratio (*w*/*w*). The complexes were prepared for three wells at the same time (but are shown on per well basis) and divided into three wells for treatments.

siRNA Conc.	Transfection Reagent /siRNA (*w*/*w*)	Transfection Volume	RPMI	siRNA	Transfection Reagent
		(μL)	(μL)	(0.14 μg/μL)	(1 μg/μL)
20 nM	6	100	98.4	0.8	0.8
40 nM			96.9	1.6	1.5
60 nM			95.3	2.4	2.3
80 nM			93.7	3.2	3.1

**Table 2 biomolecules-15-00115-t002:** Overview of statistical analysis for the data presented in Figure 5a,b.

Polymer Group		Prime-Fect					PEI1.2k-PHPA-Lin9				
**siRNA type**		FLT3	KMT2A::MLLT3	CTRL+FLT3 (1:1)	CTRL+KMT2A::MLLT3 (1:1)	FLT3+KMT2A::MLLT3 (1:1)	FLT3	KMT2A::MLLT3	CTRL+FLT3 (1:1)	CTRL+KMT2A::MLLT3 (1:1)	FLT3+KMT2A::MLLT3 (1:1)
**siRNA** **conc.**	40 nM	****	****	***	****	****	ns	ns	ns	ns	ns
	60 nM	***	***	**	*	***	**	ns	ns	ns	**
	80 nM	**	***	**	**	****	*	*	ns	*	**

ns: not significant, *: *p* ≤ 0.05, **: *p* ≤ 0.01, ***: *p* ≤ 0.001, ****: *p* ≤ 0.0001.

## Data Availability

The original contributions presented in this study are included in the article/Appendix A; further inquiries can be directed to the corresponding author.

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
