# Peer review of "Lipopolymers as the Basis of Non-Viral Delivery of Therapeutic siRNA Nanoparticles in a Leukemia (MOLM-13) Model"

_biomolecules, 2025, doi:10.3390/biom15010115_

Round 1
Reviewer 1 Report
Comments and Suggestions for Authors
In this paper Yotsomnuk et al. report a study aming at exploiting siRNA LNPs targeting two oncogenes FLT3 and KMT2A::MLLT3 involved in development of acute myeloid leukemia (AML). Two different LNP formulations are compared (a commercial (PrimeFect, and a PEI derived formulation from the literature). The experiments are primarily conducted using the (Luc(+))MOLM-13 and MV4-11 AML cell lines, but a preliminary study in an ip mouse model using the Luc(+)MOLM-13 cells and local i.p. administration of the siRNA LNPs was also performed.
The results show that in vitro cell viability and target mRNA is reduced by treatment with anti-FLT3 siRNA as well as anti-KMT2A::MLLT3 siRNA and more efficiently using a combination of the two siRNAs. These experiments were performed using the two LNP formulations in parallel, and only small differences in effect was observed. Treatment with control siRNA formulations showed significantly lower effects on cell viability. It was also demonstrated that the siRNA treatment induce cellular apoptosis as at least part of the mechanism for the decreased cell viability.
The in vivo mouse model experiments showed a small tumor cell reduction after 13 days of treatment every second day but only when using the PrimeFect, and no difference for the PEI derived formulation.
Considering that effective LNPs formulations for clinical use are available (exemplified by the drug ”Onpattro” (as well as mRNA vaccines), and the present LNPs have not been compared to these, the scientific and general importance of the present paper seems limited
Specific points
1. p2, line 67-8: ”reaching the clinical potential of siRNA therapies is still a considerable challenge”. Since there are 6 FDA approved siRNA drugs in the clinic and one of these is an LNP formulation, the statement does not seem justified!
2. P2: Indeed, it would have made sense to experimentally have compared to the above formulation in order to be at the front line.
3. P2, Line 98: ”KMT2A contributes to the initiation of transcription”: KMT2A is a histone methyltransferase, involved in epigentic control of gene expression.
4. P3: The source of the siRNA, as well their base sequence, modifications etc. is missing
5. P4, Table 1: This table needs revision: is the number in column 2: on weight or molar basis? Column 4: how did the authors measure uL with the indicated precision? column 5: How does “0.14 ug/uL” fit in? Column 6: what does the values represent
6. P 9, Figure 2a: These microscopy images are useless, they are too small, there is hardly any signal, and the resolution is poor. The intracellular localization/distribution must be clearly visible.
7. P15, line 535-6: the amount must be in ug not mg
Thorough proofreading of the manuscript should be done
Comments on the Quality of English LanguageThorough proofreading is required
Author Response
The attached file contains line-by-line response to the review comments.

Reviewer 2 Report
Comments and Suggestions for Authors
This is a well written and organized manuscript focusing on the development of a lipopolymer for the delivery of siRNA targeting leukemia model. The manuscript is addressing an important problem with a relatively novel approach. The manuscript looks good in its current form, but needs some minor improvements as outlined below.
Abstract:
- Please provide more explanation and put more emphasis on the novelty of this study. What makes this system different from other lipopolymers or other siRNA delivery systems?
Introduction:
This section is well organized and written but just couple minor points to mention:
- Please mention the potential side effects of siRNA and how these side effects can be potentially eliminated with the approach proposed.
- Please also elaborate about the limitations or challenges associated with electrostatic complexes. For instance, strong electrostatic complexation may provide high siRNA protection but diminish efficient release/dissociation at target site.
Materials and Methods:
- Details about the preparation and characterization of lipopolymers are missing. Although it was referred, please briefly describe.
- Please justify the selection of 6:1 ratio for the complexation along with the selected dose regiments.
- Please fix Table 1 caption.
- Please justify the selection of only male mice and not any female mice. Sex as a biological variable should be addressed.
Results and Discussion:
- Please fix reference [39-40] text color.
- Please elaborate the stability of the siRNA/lipopolymer complex in terms of siRNA protection against RNase enzymes and serum proteins. This can be done by simple gel electrophoresis assay. Also, provide information about the long terms storage stability of this system.
- Please provide better images for Figure 2a or increase the brightness to make them more visible.
Author Response

(The authors gave the same response as above.)

Round 2
Reviewer 1 Report
Comments and Suggestions for Authors
The manuscript has been improved, but some issues remain.
1. Table 1 is still puzzling, as it is unclear how the dilutions were (technically) made. For instance, it states that the 40nM siRNA solution is composed of 96.865 uL RPMI 1.600 uL siRNA sol. and 1.535 uL transfection reagent. How was this accuracy practically achieved?
2. The microscopy images (Figure 2) are still of very poor quality, and can only be used to estimate total fluorescence per cell, and it should be repeated for confirmation. Also, the microscopy images (Fig. 2a) should be moved to supplementary. Finally, the method for determination of quantitative cell fluorescence should be described in detail. The variation indicated in Fig. 2b seems very low compared to the images
3. The quantitation of the in vivo data (Figure 8) is difficult to understand. In all cases the change in radiance from day 09 to day 13 is around 10 fold (Figs c, d), whereas the images themselves (which I admit are not directly quantifiable by eye) do not appear vastly different. What is the explanation for this?
Author Response
We thank the reviewer again for inspecting our submission. Below are the new review comments and our response.
- Table 1 is still puzzling, as it is unclear how the dilutions were (technically) made. For instance, it states that the 40nM siRNA solution is composed of 96.865 uL RPMI 1.600 uL siRNA sol. and 1.535 uL transfection reagent. How was this accuracy practically achieved?
We work with pre-determined stock solutions based on the manufacturer supplied siRNA. We subsequently rely on calibrated (calibration done in our lab every 6 months or so) eppendorf pipetters in order to dispense and dilute the right volumes of the materials to obtained the desired concentration. We do not employ additional assays to confirm the concentrations, having confidence of the original concentrations associated with stock solutions from the manufacturer. This is standard practice in the field. But if the reviewer is referring to the 3 decimal point shown for the volumes, this must have been a mistake in the submission of the manuscript - our pipetters can hand up to 0.1 uL as the smallest increment. We now fixed this excess d.p. shown.
- The microscopy images (Figure 2) are still of very poor quality, and can only be used to estimate total fluorescence per cell, and it should be repeated for confirmation. Also, the microscopy images (Fig. 2a) should be moved to supplementary. Finally, the method for determination of quantitative cell fluorescence should be described in detail. The variation indicated in Fig. 2b seems very low compared to the images
As we explained in our previous response, these images are intended as a visual confirmation. Nothing more. A general reader will see the differences in fluorescence and confirm the numbers obtained from the flow cytometry. We note that the 2nd reviewer had raised this issue as well and he/she was satisfied with our response. As suggested by the reviewer, we moved the images to Supplementary Data. The flow cytometry analysis was described in almost all of our previous papers on this topic and it is quite a precise method to assess fluorescence associated with cells (so low error bars are not surprising to us, since it is a very reproducible method and we have perfected cell processing needed for this assay in the last 15 years). We provided additional information as requested by the reviewer on this method in the results section, explaining the significance of the specific parameters measured.
- The quantitation of the in vivo data (Figure 8) is difficult to understand. In all cases the change in radiance from day 09 to day 13 is around 10 fold (Figs c, d), whereas the images themselves (which I admit are not directly quantifiable by eye) do not appear vastly different. What is the explanation for this?
We do not have a strong explanation for this issue. It is clear that the eye has only a limited capacity to difference between qualitative images taken in this study (resolution of our eyes is not good enough). But when we quantitate the obtained images for luminescence values using the detectors associated with the animal imager, statistically different values emerge among the study groups. We previously mentioned in the submission that the response obtained was not over-whelming with our therapy so that the differences are not overly obvious. But the employed statistics do tell us about the difference being significant and accordingly we are making relatively conservative conclusions in our submission. We considered our statement sufficient in this regard in the submission and, hence, we took no action on this matter.
Reviewer 2 Report
Comments and Suggestions for Authors
Thank you for the responses.
Author Response
It appears that this reviewer is satisfied with our response since there were no additional comments visible to us. We appreciate the reviewer's time and effort in inspecting our submission.
Round 3
Reviewer 1 Report
Comments and Suggestions for Authors
The manuscript has been improved. However, a few issues still remain:
Table 1: Column 4 (RPMI): the volume is still given with 3 sub uL digets! And in columns 5 & 6, the numbers are in uL (this should be stated in the Table).
P9, line 395: unlabeled siRNA is not negative control, it is background. A negative control would be e.g. non-formulated siRNA.
P16: The authors in their response agree that a significant quantitative discrepancy seems to exist between the image scans and the electronic quantification (Fig. 8), and provide no convincing explanation for this. Since this data is the most critical for evaluation of the significance of the overall importance of the study, this apparent discrepancy must be properly addressed/resolved.
I assume that the authors can confirm that the scanner settings were identical for all mice for all three scannings (day 05, 09 and 13)?
I also assume that the mice were individually identifiable, allowing to follow the same mouse through the three time points?
Author Response
Below are comments from one reviewer (in black text) and our response (red text). We again appreciate the reviewer's efforts to inspect our submission and did our best to address his/her concerns:
Table 1: Column 4 (RPMI): the volume is still given with 3 sub uL digets! And in columns 5 & 6, the numbers are in uL (this should be stated in the Table). This is now fixed.
P9, line 395: unlabeled siRNA is not negative control, it is background. A negative control would be e.g. non-formulated siRNA. Note that line 392 uses the word 'negative control'. We changed the wording to background in line with the reviewer's suggestion.
P16: The authors in their response agree that a significant quantitative discrepancy seems to exist between the image scans and the electronic quantification (Fig. 8), and provide no convincing explanation for this. Since this data is the most critical for evaluation of the significance of the overall importance of the study, this apparent discrepancy must be properly addressed/resolved. This issue was not brought up before and we do not agree that there is such an obvious discrepancy. It is correct that we have a difficult time seeing 'subtle' differences in images among the study groups at a specific time point (due to limitations of our eyes to discriminate visual information) unless there are glaring differences. This is typical of any studies that utilize this type of analysis. However, quantitation is done without bias and we do see come differences in quantitative values. We note that the differences are not major, but they are significant. This is also stated in the manuscript as follows: "Despite its significance, the inhibition of growth was relatively small, unlike our other studies which investigated growth of local tumors [41], where >50% inhibition of growth was evident within the study periods." The reader should be aware of the magnitude of the effect observed as a result of our statement. Hence, no action was taken on this issue.
I assume that the authors can confirm that the scanner settings were identical for all mice for all three scannings (day 05, 09 and 13)? We confirm that the scanner settings were identical for all mice and at different time points.
I also assume that the mice were individually identifiable, allowing to follow the same mouse through the three time points? Yes. By means of ear tags, we followed the Luc activity in individual mice as a function of time.
Round 4
Reviewer 1 Report
Comments and Suggestions for Authors
The concerns have been adequately addressed.